**Data Availability Statement:** All relevant data are within the paper.

**Funding:** The authors received no specific funding for this work

# A qualitative insight into researchers' perceptions of gender inequality in medical and dental research institutions in Nigeria

**Morenike Oluwatoyin Folayan**[1,2]*, **Adekemi Olowokeere**[3], **Joanne Lusher**[4], **Olabisi Aina**[5], **Ana Gascon**[1], **Guillermo Z. Martínez-Pérez**[1]

1 Department of Physical Sports, Faculty of Health Sciences, University of Zaragoza, Zaragoza, Spain, 2 Department of Child Dental Health, College of Health Sciences, Obafemi Awolowo University, Ile-Ife, Nigeria, 3 Department of Nursing, College of Health Sciences, Obafemi Awolowo University, Ile-Ife, Nigeria, 4 Provost's Group, Regent's University London, London, United Kingdom, 5 Department of Sociology, Faculty of Social Science, Obafemi Awolowo University, Ile-Ife, Nigeria

* toyinukpong@yahoo.co.uk

## Abstract

### Objective

The aim of the study was to gain a qualitative insight into scientific researchers' perceptions of gender inequality inside Nigerian research institutions through an investigation of how gender equality is enacted in medical and dental research institutions in Nigeria.

### Methods

This descriptive and cross-sectional qualitative study probed decision-making around navigating gender inequity and explored opinions about how a supportive environment for female medical and dental researchers could be established. Data were collected through semi-structured telephone interviews with 54 scientific researchers across 17 medical and dental academic institutions in Nigeria between March and July 2022. Data were transcribed verbatim and analyzed using thematic analysis.

### Results

Three core themes emerged: institutionalized male dominance in research institutions; changing narratives on gender equalities in research and academic enterprise; and women driving the conscience for change in research institutions. Female medical and dental researchers' perceived gender equality was challenging mainstream androcentric values in knowledge production within the medical and dental field; and queries the entrenchment of patriarchal values that promote a low number of female medical and dental trainees, fewer female research outputs, and few women in senior/managerial positions in the medical fields.

**Competing interests:** The authors have declared that no competing interests exist.

## Conclusion

Despite the general view that change is occurring, a great deal remains to be done to facilitate the creation of a supportive environment for female medical and dental researchers in Nigeria.

## Introduction

Gender equality is a visionary pursuit that carries an implication that economic, social, and cultural attributes and opportunities associated with different genders by different society should not confer difference in expectations that debar pursuits and aspirations, can positively influence individual and social development [1]. Moreover, equality is critical for the socioeconomic stability of countries as it promotes and guarantees peace and social justice. Three areas in which gender equality can foster progress is in science, education, and health [2]. Medical and dental researchers stand at the fulcrum of social development through their engagement in these three domains [3]. They also contribute to economic development through their work on understanding disease and promotion of medicine, vaccines, diagnostics, and effective public health messages. The work of scientists in research institutions often requires training others to advance science to support disease eradication and quality of life. This opportunity opened to medical and dental researchers through the promotion of gender equality has yet to be optimized.

In Africa, female researchers face greater challenges in the medical and dental health contexts limiting their ability to make optimal contributions to individuals and society [4]. For example, in West Africa, medical and dental researchers are faced with gender values and norms that assign women to domestic tasks and responsibilities that reduce the time they can dedicate to research [4]. This may explain why women spend more time teaching and less time researching, when compared with men [5]. Additionally, gender-blind organizational culture and institutional policies make it difficult for women to attain leadership positions and place them at risk of low opportunity for participation in science [4, 5]. Moreover, female medical and dental health researchers can become distracted from investing in the process of challenging the gender-blind systems by dedicating attention to preserving relationships with their male spouses [4].

Like many other gender-blind research institutions in and outside of Nigeria, including medical research institutions, men in research have a greater number and quality of research outputs than women who are in research [6–8]; and fewer women are seen in top-office in research institutions [9]. The tendency for this is high because of the large number of male medical and dental students [10, 11]. This is also driven by the high number of male students who pick a pre-college science track in secondary school [12]. The high number of men in medical and dental research institutions and the high number of men in managerial posts reflects a complex sociological phenomenon that enshrines these patriarchal practices through the use of gender exclusionary strategies that maintain the male monopoly [13, 14].

There is limited information on the gender distribution of researchers in Nigeria. The available evidence indicates that the number of women in dental institutions had steadily increased from 36.2% in 2003, to 42.5% in 2013. Thus, the predominance of intake of men remains [15]. There are clearly gender differences in fields of specialization with a steady increase in the number of women in leadership positions [15]. However, this increase in female participation in dental academia may not quite reflect medical practice. For example, Ogunbodede [16] indicated an observed discrepancy in the increase in number of practicing female dentists

versus practicing female medical practitioners over a 10-year period. While the number of dentists increased from 15% to 35% between 1981 and 2000, the increase for medical practitioners only shifted from 15% to 19%. Similarly, the number of male doctors in Nigeria is consistent with roughly twice the number between 2017 and 2019 [17]. Although male dentists do not double the number of female dentists, the number of male dentists (810) outweighs female dentists (555) [18].

A gender equality gap could not, however, be determined purely on grounds of numbers. This would be time-bound and to the continued detriment of developing an equal society. There is also no guarantee that a reduction in the number-gap would change the current paradigm. Prior studies have indicated that having more women educated in science not will change the status quo of more men holding senior positions [19–22]. The envisioned change in gender representation in the research institutions and research managerial positions needs to be driven by the collective concern and commitment to improving the quality of research outputs for the health and wellbeing of the society through the participation of women [23].

The theoretical perspective that informed the design of this study therefore, used a feminist institutionalism analytical lens that would enable the exploration of gendered institutions and their gendering effect [24]. The feminist institutionalism analytical approach enhances the exploration of the gendered dimensions of structures of power and behavior and the role played by institutional informal structures, processes, values, and norms. [25, 26]. It enables an analysis of how informal institutions interacts with the formal systems; through roles played by gendered rules, actors, and outcomes, to produce gendered outcomes. Also, the feminist institutionalism analytical approach provides a theoretical lens that allows for gendered power relations and the processes that makes such relations visible [25].

Despite it being apparent that gender inequality practices exist among professions, little is known about how female medical and dental health researchers use the potentially transformative opportunities that do come their way. Are there career trajectories that make it possible for female medical and dental health researchers to access and maximize their use of these transformative opportunities in education and career development? Does the cultural context of female medical and dental health researchers support the institutionalization of gender inequality in ways that limits their ability to facilitate gender equality in their profession?

The present study aimed to address these research questions in order to provide insight into the career advancements of medical and dental academic health researchers irrespective of their gender. This qualitative study investigated how gender equality is enacted in the medical and dental research field; and explored male and female researchers' perceptions of gender inequalities in medical and dental research institutions in Nigeria. It probed decision-making around navigating gender inequity within research institutions and explored opinions on how a supportive environment for female medical and dental researchers in Nigeria could be established.

## Materials and methods

This study adopted an academic literacies perspective that accounts for context, culture, and genre [27, 28]. The theoretical framework applied in this study was the preference theory due to its appropriateness for exploring researchers' investment in efforts to mainstream gender considerations in institutional processes; whilst recognizing the need for women to simultaneously meet family and work responsibilities [29].

### Study design, study site and study participants

This study formed part of a larger qualitative study that was conducted in Nigeria to determine barriers and ways to resolve gender equality in medical and dental research institutions in

Nigeria. Female and male faculty members of 17 universities in Nigeria took part in this research study between March and July 2022. Participants were able to read and communicate in the English language and defined themselves as academics in either the health, medical, or dental education field because they promote, design, conduct and disseminate biomedical, clinical, and socio-epidemiological research in Nigeria. All participants resided in Nigeria, were adult members of academic or research institutions working on health issues and consented to take part in a one-hour interview.

## Sample size

It was estimated that three study participants would be recruited from each of the 17 institutions that hosted a medical and dental school in Nigeria. Therefore, 54 interview slots were allocated equitably amongst professors, readers, senior lecturers, and lecturers (the entire spectrum of designations in the academia in Nigeria). The slots were also divided equitably among dentists and medical personnel, and in a proportion of 2:1 for female: male interviewees. These designates were then randomly allocated to each institution as indicated in the S1 Table. This sample size was adjudged to be adequate to generate rich information; and allow for saturation to occur. With a non-response rate of 20%, it was anticipated that the final sample size for the in-depth interview will be 43. Saturation is often reached with a sample size of 12 persons when working with a homogeneous group like that for this study [30].

## Sampling procedure

A purposive and convenience sampling was used to identify potential participants working in a medical and dental health academic institution and conducting research in that institution. The diversity of respondents was ensured by recruiting study participants from all the academic cadres in the medical and dental institutions.

Study participants were recruited through a combination of purposive sampling and snowballing. Peers of the principal investigator in each of the 17 institutions were contacted and asked to identify a possible respondent that fit the profile of respondent to be interviewed in their institution. If the interviewee met the inclusion criteria, the principal investigator contacted her/him by telephone or via Email/WhatsApp. The purpose and objectives of this study was explained, the interviewee was invited to take part in an in-depth interview and a date was scheduled for the interview. Before the scheduled date, written informed consent was sought. When the consent form was filled and sent back, the interviewee was then enrolled as a study participant.

At the end of each interview, participants were asked to share the name of a colleague who may be interested in the interview. That colleague was then contacted and the process for enrolment was repeated until the target number of participants had been reached. Whenever a study participant refused study participation, the participant was replaced by an eligible study participant in the pool of contacted researchers.

## Study procedure

An interview schedule was adapted based on a focus group discussion held with a convenience sample of 12 researchers working in medical and dental academic institutions. These were six male and six female researchers from Belgium, Brazil, Malaysia, Iran, Nigeria, United States of America, and Turkey. These researchers had a history of working in the West Africa region. The expert consultation was held via a conference call in June 2021 and aimed to explore perspectives on the scope of intervention carried out by university faculty members on gender equality in education and professional development. Before the discussion took place,

participants received a one-page concept note about the main study, which included a brief description of the conceptual framework for the study; the working definitions of gender equality (people of all genders have equal rights, responsibilities and opportunities); and the aim of the discussion. The outcome of this discussion formed the basis of the interview schedule used in the main study.

In-depth interviews were conducted between March and July 2022 during the COVI-19 pandemic. Interviews were conducted using Telegram and WhatsApp video calls to identify interviewees perspectives on observed gender differences in career progression and trajectory of medical and dental researchers in Nigeria; as well as gender-related barriers to opportunities for changing this trajectory for future female researchers. Interviews were conducted between March and July 2022. All interviews were conducted in English and audio-recorded. Interviewees were required to switch on their video to enable the interviewer ensure the interview was being conducted in privacy. Notes were taken during the interview. The time range for the interview was 26 minutes to 71 minutes with a mean time of 55 minutes.

## Data analysis

The purpose of data collection was to seek richness of information and to saturate all concepts and categories emerging from the in-depth interviews. After each interview, transcripts were transcribed verbatim, and were read and reread to reveal emergent themes. Table 1 presents ten broad topics that were explored through these interviews. Interviews were transcribed verbatim into a password protected Microsoft Word document accessible only via a single password protected computer. Anonymized transcripts of the recordings were checked to verify their accuracy and completeness compared to the audio recordings. Personal identifiers and names of places and institution were not transcribed.

Transcribed interview recordings were imported into Atlas.Ti and read and re-read to identify codes and categories using an inductive approach to code, analyze and report on [31, 32]. This process helped gain familiarity with the data and achieve new insights by analyzing for recurring themes and issues that represented answers to the questions; and to draw conclusions from the responses.

A codebook was inductively developed from themes that had been generated and from analytical questions intended to elicit a thorough, nuanced exploration of gender equality in medical and dental research. Coding and analysis were led by the first author of this report. A second qualitative researcher was consulted for extra coding to ensure inter-coder reliability during the process. The adoption of this particular approach ensured the identification and description of new codes and subthemes within the transcripts and this procedure continued until the point of saturation was reached.

During this coding process, novel codes that emerged from the data were included to review the initial generated codebook. The transcripts were again re-read using the new codebook. This approach was employed to develop categories, which were then explored and used when discussing the pre-conceived topics. The concepts and categories of analysis were defined using the words of the participants. Data were organized into key themes and subthemes generated by the coding process, and excerpts and illustrative quotes of general insights and of deviant cases from the transcripts were selected to substantiate the presentation of the key findings in this report. The informants' own words were also used to report the findings. Attention was paid to the researchers' reflexive journals to ensure that informant biases were not introduced. The Consolidated criteria for Reporting Qualitative research guidelines were considered.

**Table 1. Ten broad topics included in the discussion guide.**

| | |
|---|---|
| Opinion on gender equality | What do you understand by gender equality? What are your thoughts about gender equality in research institutions? What do you think about gender equality in medical or dental health research area? |
| Opportunities related to the sex/gender of the researcher | What are the opportunities you've encountered during your medical or dental health research career? *Prompt about gender/sex-related opportunities* . . .do you think medical or dental researchers of the opposite sex encounter the same type of opportunities? |
| Sex and gender related research obstacles | What are the obstacles you've encountered during your medical or dental health research career? *Prompt about sex and gender related obstacles* . . .do you think medical or dental researchers of the opposite sex encounter the same type of obstacles? |
| Difficulties female researchers face in their professional career | What are the difficulties and challenges female researchers face to . . .receive an education as researchers? . . .lead their own research projects? . . .access funding, grants, scholarships? . . .publish? . . .combine academic and research work? . . .reach top management positions? In comparison with men: How are all these things similar or different? |
| Integration of 'gender perspective' or 'gender lens', 'gender approaches' in medical/dental research | Perception. . . 1. In their own research agenda     In the medical/dental research agenda of colleagues in     the same field |
| Professional experience with sex and gender mainstreaming in health research conduct | How have you practiced sex and gender mainstreaming in medical or dental health research? *Probe, what made it (not) possible; what where the challenges faced; what are the barriers created by knowledge, skills and access to resources; how were these barriers and challenges addressed; if they were not addressed, what made it challenging; has the societal perception of women had any significant influence; if yes/ no, how? What spurred you to take such initiative.* |
| Recommendations for gender equality in research and academia | Can you give us recommendations on how to promote and institutionalize gender equality in research practices and in the academia? |
| Opinion of researcher regarding sex and gender mainstreaming in medical or dental health research conduct | We are going to talk about integrating gender and sex in medical or dental health research: What in your opinion are the opportunities there are with respect to giving considerations to sex and gender in health research planning and implementation? *Probe for opportunities that can be derived from the institution and from peers* What changes can result from integrating gender and sex into medical or dental health research practices? (Advantages disadvantages) |
| Female medical and dental health academic researchers' sorority and solidarity efforts in support of the elimination of gender inequality practices | How well have female colleagues been supported to succeed with their career as medical or dental health researcher? *Probe on the career progression pathway in the institution; how junior female colleagues have fared; what senior colleagues have done/can do to improve career progression of early career females; perception on how gender/sex dynamics have impacted on institutional and individual support; and what can be done to improve the current institutional and individual systems and practice to support early career medical and dental academia.* How do you feel about men joining women's efforts and initiatives to improve gender equality in medical dental research? |

*(Continued)*

**Table 1.** (Continued)

| Recommendations for better sex and gender mainstreaming in medical or dental health research conduct | Can you give us recommendations for better integration of gender and sex in medical and dental health research in West Africa?<br>How feasible would it be to implement the recommendations you're suggesting?<br>What would be necessary to convince women and men in medical dental research to pool time resources efforts to make reality all the transformations you suggest. . . |

## Ethics

Ethics approval was obtained from the Institute of Public Health, Obafemi Awolowo University, Ile-Ife, Nigeria (IPH/OAU/12/1617). All informants signed an informed consent form.

# Results

## Participants' characteristics

Table 2 presents data on the sociodemographic profile of the 54 female and male medical and dental professionals who participated in the in-depth interviews. The sample presents 48% females and 52% males, with their ages ranging from 33 to 62 years (what of the age mean?). Participants were educated to postgraduate level and most of whom were married. The number of children of interviewees ranged from 1 to 5.

**Table 2. Sociodemographic profile of participants (N = 54).**

| Characteristic | Male (n = 28) | Female (n = 26) |
|---|---|---|
| **Age** | | |
| 20–40 | 3 | 6 |
| 41–60 | 24 | 20 |
| 61–70 | 1 | - |
| **Marital status** | | |
| Single | - | 2 |
| Married | 28 | 24 |
| **Profession** | | |
| Dentists | 11 | 10 |
| Medicine | 8 | 5 |
| Surgery | 5 | 3 |
| Basic sciences | 4 | 8 |
| **Designation** | | |
| Professor | 3 | 3 |
| Associate Professor | 4 | 2 |
| Senior Lecturer | 16 | 13 |
| Lecturer I | 3 | 6 |
| Lecturer II | 2 | 2 |
| **Number of children** | | |
| 0 | 5 | 9 |
| 1–4 | 16 | 16 |
| 5 | 7 | 1 |

## Profile of refusals

An initial 44 (29 medical and 15 dental researchers; 30 females and 14 males) contacts were made. Of these, 12 (7 medical and 5 dental researchers; 10 women and 2 men) did not respond to contact made. One did not meet eligibility criteria, and eight (7 medical and 1 dental researchers; 4 females and 4 males) declined participation. The 21 consented respondents helped reach other participants through the snowballing process

## Emergent themes

Three core themes emerged from the data that each reflected participants' perceptions on how female medical and dental researchers' make decisions to navigate the constraints within the research institutions in which they work; and how they act to promote a supportive environment for their female peers. These themes were: (1) Ingrained patterns of institutionalized male dominance in research institutions; (2) Hopes for a changing narrative on gender equalities in research; and (3) Women driving the conscience for change. These themes, along with the subthemes and extracts are presented in Table 3.

**Ingrained patterns of institutionalized male dominance in research institutions.** Women felt under-represented and men overrepresented in most fields in the medical and dental fields. Women expressed an opinion that the medical and dental fields are male-dominated and that women need to compete more with lower status positions, as one participant expressed:

> 'If you go to male -female enrollment in school, you will find out that the percentage is higher for males than females. Then, when you even come to the university, at least I can assert a guess that in my place, if we are like 35 doctors there will be like 7 females' **Dentist_male**

Interviewees of both genders perceived that the number of women in the medical and dental profession is increasing, though they remain underrepresented in managerial positions in universities in Nigeria and are less likely to be promoted or elected to managerial positions. Women identified that they had had to put in a lot of effort to demonstrate that they were capable of doing just as well as or even better than men as indicated by the following extract:

> It was not a small battle to convince learned people like professors, medical doctors saying a female can do this. Even up until now, we are yet to have our female first Vice Chancellor.' **Basic Sciences_Male**

**Table 3. Emergent themes, subthemes, and illustrative quotes.**

| Key themes | Sub themes | Illustrative quotes |
|---|---|---|
| **Ingrained patterns of institutionalized male dominance in research institutions** | • Underrepresentation of women in medicine and dental specialties<br>• Underrepresentation of women in managerial positions<br>Institutionalization of patriarchy and androcentric values | *There are far more male medical students than female medical students. Things are even worse in Northern region of Nigeria where male children are privileged. I think that's the foundation of the observed skewness in research. More women need to get an education to have greater opportunities in research* |
| **Hopes for a changing narrative on gender inequalities in research** | • Increased public discussion<br>Speaking and 'acting up' for gender equality | *The present Provost of the College is female. Previously, they were males, I think now, we are beginning to have females. I think in the area of research, nobody is going to marginalize a female who has an idea* |
| **Women driving conscience for change** | • Female gender bias in grant opportunities<br>Gender mainstreaming into research systems | *I've actually seen some grant and training calls specifically for female researchers. But at the same time that you need to dedicate to these calls are difficult to make out as a female at the age when you are eligible to apply. For example, when you are pregnant or you're breastfeeding, that's like 18 months of whatever program. At this time, you won't really be able to put in your best* |

As per the participants' opinions, gender inequality results in uneven power relations, entitlements, social values, responsibilities, and duties in patriarchal societies. The socialization process also affects how each gender perceives oneself and the power and influence they have. The majority of female researchers observed a male dominant culture in the medical and dental profession, which not only limits the opportunities for selection or nomination into leadership position, but also medical and dental specialization opportunities. As identified, men are perceived by female participants as afraid of women altering the status quo in academic and medical research:

*'Because of socialization the moment you are born, your parents tell you how to conform to gender expectations. Females are socialized not to do things that are tedious. This influences even professional. You see females being discouraged from being a surgeon and encouraged to be things like pediatric dentist, dentist.'* **Medicine_Male**

**Hopes for a changing narrative on gender inequality in research.** Participants of both genders voiced increasing public discussions on gender, gender equality, gender bias and discrimination; that is making it possible for a gradual shift in gender-biased practices in the medical and dental fields. These public discussions for change are happening by female professionals who speak up and advocate for evidence-based changes to gendered practice. The active drives for gender equality in the medical and dental fields is resulting in the rising enrolment of women in medical and dental schools, despite enrolment still being largely dominated by men. Participants opined that there is a growing awareness of opportunities for women to pursue careers in medicine and dentistry:

*'I will say that I think we have more of male researchers in medical and dental, but I think the trend is changing, there is also a male dominance in leading research but that is also changing because I think there is a lot of emphasis now on balancing the gender composition of researchers and also gender balance in recruitment. I think people are now thinking in that direction but before most of the research in medical and dental field include more of male than female as researchers and study participants.* **Dentist Male**

There were testimonies of three female medical and dental professionals being the first to hold key managerial positions. For these women, breaking the gender barrier was an effort to create the needed pathway to make it easier for other women to come on board:

*'I was the first female consultant in the department, and the first female professor in the department'. You know most at times, if you are able to cross the first few hurdles, the rest becomes easier. So, maybe I'm the sacrificial lamb of the department.* **Surgeon_Female**

**Women driving conscience for change.** Participants of both genders identified that the selection criteria for many grant opportunities were biased towards women; a number of male participants felt this skew opportunity for women gave women advantages. These biased opportunities were efforts by the granting agencies to drive gender equality in the medical and dental research fields. Male and female respondents, however, opined that the opportunities were not gender biased, while one female researcher commented that the opportunities open to women are not real opportunities, as women are often not able to make the best of these opportunities:

*'. . .an organization that gave gender differences in the cut-off age for application of grants– the cut-off age was lower for males than females. This was because the granting agency recognized that females start a lot later than males in their research career trajectory because of their social responsibility of caring for the babies and other unpaid care duties.'*
**Medicine_Female**

Other opportunities for female researchers identified were gender mainstreaming into the composition of research teams not only for gender equality but also to improve the quality of the research outcomes, as diverse perspectives enhance the quality of the design and implementation of research. Gender mainstreaming was identified as important for many reasons, one of which for institutional building. Also, participants identified the need to build the capacity of women to be competitive, and for gender equality advocacy and sensitization of gender-blind research institutions.

To address barriers that prevent women from gaining access to tertiary education, professional research opportunities and promotion at the same rate as men. Few female and male participants identified the need for gender-sensitive policies that mainstream gender considerations in the appointments, recruitments, selection process of female medical and dental professionals into leadership roles; gender considerations in the access to grant opportunities; and opportunities for senior female mentoring of early career (female) researchers. Such policy drives and change can be achieved through the collaborative efforts of female medical and dental professionals. A few participants in this study proposed that gender study centers should be established within medical and dental institutions; and they be saddled to handle gender related issues:

*'The establishment of gender centers will probably promote gender equality generally. They can also generate research-based evidence that can address the 'why', 'how' and the value added by promoting gender equality.'* **Medicine_female**

Overall, participants in this study argued for institutional policies that help to drive gender sensitivity. Data pointed to policies needing to promote gender equity at the managerial and administrative levels while focusing on skills and expertise. The opportunities should be open equitably to everybody.

## Discussion

The current study identified a male dominance in the research outputs of medical and dental researchers in research institutions in Nigeria. This male dominance also reflects in the inequitable distribution of managerial positions of the institutions. The gendered operations of medical and dental research institutions is driven by the absorption of the societal patriarchal values. Individuals in this study expressed a paradigm shift driven by individual and collective bodies of women in academia driving a conscience for change. Other opportunities identified to drive the change process included enacting institutional policies that promote gender equality; establishing gender focused units in research institutions dedicated to implementing these policies; continued advocacy and awareness creation for the change to happen; mentorship by women for women and for men; and building the capacity for women to actively engage with others in the research enterprise.

A benefit of these findings is that they provide a contextual and rich foundation of evidence that supports prior research on the inequitable representation of women [4, 33, 34]. Moreover,

a focus on medical and dental research institutions has allowed a deeper exploration of contextual professional factors that may promote gender inequality in a research setting.

Indeed, participants in this study perceived gender inequality as enacted through institutionalization of societal patriarchal and androcentric values that may make domestic responsibilities and career breaks for domestic reasons have far more reaching impact on women's research outputs, and career progression; compared to their male counterparts. It is likely that poor environmental support for research in Nigerian institutions have more impact on women than men who are less able to access sponsored opportunities for capacity development due to the need to stay home even when these opportunities are presented [35]. For the same reasons, women may be less able to take up research grant opportunities even when grants are biased towards the selection of women, because of the care responsibilities they are encumbered with. These distractions from capacity building and empowerment opportunities during early career development years are challenging to catch up with in later years of a woman's career, which thereby increases the gender competency gap. The failure to adjudge years of home management as human managerial skills, and poor accounting of home care as work skills continue to make women fall behind in the ratings for skills to handle managerial offices.

Though institutional policies and advocacies can help to bridge these gaps, they are unlikely to be eliminated. Gender equality policies are challenging to implement, but when implemented, significant progress can be made with gender mainstreaming [36]. Gender-sensitive institutional policies in medical and dental research institutions, implemented by established gender focal units, may help to drive the shift towards gender equality in research outputs and numbers of female appointments into senior cadres. These policies will need to promote a gender sensitive review of assessment criteria for appointment and promotion. Further research is necessary to better understand how home-management skills can be rated, groomed and adapted as administrative skills. Efforts in these directions may help to eliminate the managerial position gender equality gap. This may also facilitate men in taking on home care roles in the knowing that they will not be worse-off for doing so.

Furthermore, participants' voices pointed to continued advocacy and awareness creation. One of the roles of the Medical Women's Association of Nigeria is to advocate for favorable policies for women, and they have done so successfully for many issues related to clinical practice [37]. One of which is for paid maternity leave. They have, however, achieved little in driving equality in the field of research. Women in academia may need to form pressure groups to address the issues peculiar to their needs. Pressure groups also need to partner and engage with men to promote gender equality; and advocate for new masculinities and for human rights. Advocacy seeks to narrow the gap between what is known to be effective, acceptable, and efficient and what is practiced [38]. It involves a combination of individual and social actions designed to gain political commitment, social acceptance, and system support for a particular goal or program. Though it is an effective strategy for producing policy change, it can be difficult and complex for those with limited power and resources [38]. Future work is vital for understanding how gender equality in research institutions has contributed to the attainment of the sustainable goal more generally.

While advocacy may bring about change, slowly, the mentorship of women by women and men allies in the gender equality fight could bring about substantial change in the research context [39]. Female mentors promote aspirations of other females to pursue the same career pathways through a feeling of belonging and confidence. Participants in this study reported views on mentoring actions, though, as such, are unclearly defined. The suggested efforts of reaching out to other women in medical and dental research, by those who explained that they have made it to more senior positions, can be institutionalized by research organizations, or bodies, of female professionals. The mentorship process could also facilitate building the

capacity of women to actively engage with others in research enterprise. However, mentorship is a non-formal educational system that should not replicate social norms, dominant values, or drivers that could otherwise entrench inequality and disempowerment of women by reproducing existing hierarchies and exclusions [40]. Training mentors on gender-sensitive mentorship strategies may help to avoid these possible risks.

One of the strengths of the study was the recruitment of study participants from Northern and Southern Nigeria thereby reflecting the views of male and female researchers from diverse cultural context in Nigeria. The study findings are therefore potentially comparable across research institutions in Nigeria. The study is, however, not without limitations. The data collected were limited to the perception of gender inequality in medical and dental research institutions and the coding and analysis of these data were conducted within this context. Additional themes and perspectives can be derived from the content-rich narratives of the participants; and this warrants further exploration as issues surrounding gender and cultural differences in light of gender equality in medical and dental research institutions are not fully understood. Differences in the perspectives of dental and medical researchers could also be examined separately, as the experiences of these two groups may differ.

Despite these potential limitations, results from the present study do provide insights that support a feminist institutionalist perspective that societal inequality is reproduced in political and social institutions such as higher institutions of learning [41]. Understanding how context specific institutional rules, processes and norms drives the enactment of gender inequality can help with the reform and improvement of institutional gender equality programs and strategies. This study is the first study to explore how and why gender inequality is enactment in medical and dental schools in Nigeria; and therefore, provides a framework to support possible gender reforms in these institutions.

In conclusion, medical and dental researchers perceive gender inequality as enacted in medical and dental research institutions in Nigeria through the entrenchment of societal, cultural and religious patriarchal values. These patriarchal values promote the low numbers of female medical and dental trainees, lower research outputs for female researchers when compared to that of male researchers, and fewer women in senior managerial positions. A lot still needs to be done to facilitate the creation of a supportive environment for female medical and dental researchers in Nigeria. This includes the development, monitoring and enforcing of newly created norms that assist in creating the needed support for gender equality. There is a necessity to establish a critical mass of gender experts in medical and dental research institutions who can design and promote effective mechanisms to promote gender equality practices in Nigeria.

## Supporting information

**S1 Table. Distribution of proposed 54 interviews by gender, rank and profession.**
(DOCX)

## Acknowledgments

The authors acknowledge the contributions of the time and efforts of the participants to this study.

## Author Contributions

**Conceptualization:** Morenike Oluwatoyin Folayan, Ana Gascon, Guillermo Z. Martínez-Pérez.

**Data curation:** Morenike Oluwatoyin Folayan, Adekemi Olowokeere.

**Formal analysis:** Morenike Oluwatoyin Folayan.

**Funding acquisition:** Guillermo Z. Martínez-Pérez.

**Investigation:** Morenike Oluwatoyin Folayan, Adekemi Olowokeere.

**Methodology:** Morenike Oluwatoyin Folayan, Guillermo Z. Martínez-Pérez.

**Project administration:** Morenike Oluwatoyin Folayan, Guillermo Z. Martínez-Pérez.

**Supervision:** Olabisi Aina, Ana Gascon, Guillermo Z. Martínez-Pérez.

**Validation:** Guillermo Z. Martínez-Pérez.

**Writing – original draft:** Morenike Oluwatoyin Folayan.

**Writing – review & editing:** Morenike Oluwatoyin Folayan, Adekemi Olowokeere, Joanne Lusher, Olabisi Aina, Ana Gascon, Guillermo Z. Martínez-Pérez.

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
