## [Decision Letter · Decision Letter 0]

3 Nov 2022

PONE-D-22-27154A qualitative insight into researcher’s perceptions of gender inequality in medical and dental research institutions in Nigeria.PLOS ONE

Dear Dr. Folayan,

Thank you for submitting your manuscript to PLOS ONE. After careful consideration, we feel that it has merit but does not fully meet PLOS ONE’s publication criteria as it currently stands. Therefore, we invite you to submit a revised version of the manuscript that addresses the points raised during the review process.

We look forward to receiving your revised manuscript.

Kind regards,

Claudia Noemi González Brambila, Ph.D.

Academic Editor

PLOS ONE

**Journal Requirements:**

2. We note that the grant information you provided in the ‘Funding Information’ and ‘Financial Disclosure’ sections do not match. When you resubmit, please ensure that you provide the correct grant numbers for the awards you received for your study in the ‘Funding Information’ section."

"MOF received funding to conduct the data collection and analysis"

"The authors acknowledge afworo that provided funding support for the data collection and analysis of this piece of work."

"MOF received funding to conduct the data collection and analysis"

Reviewers' comments:

Reviewer's Responses to Questions

**Comments to the Author**

1. Is the manuscript technically sound, and do the data support the conclusions?

Reviewer #1: Partly

Reviewer #2: Yes

2. Has the statistical analysis been performed appropriately and rigorously? 

Reviewer #1: N/A

Reviewer #2: N/A

3. Have the authors made all data underlying the findings in their manuscript fully available?

Reviewer #1: Yes

Reviewer #2: Yes

4. Is the manuscript presented in an intelligible fashion and written in standard English?

Reviewer #1: Yes

Reviewer #2: Yes

5. Review Comments to the Author

Reviewer #1: • Add more papers from 2017 to 2022 in the introduction section. It should be motivated for the reader and also add unique contribution.

• Address theoretical background of the study and also add theoretical implication (Support with theories).

• Author can mention objective of the study and research gap separately which enhance quality of paper.

• Add theoretical and managerial implications

• Explain conclusion, limitation and future scope separately.

• The language of the paper needs a careful editing.

• References must follow the style as per journal

Reviewer #2: Paper looks good and I strongly believe that this paper is contributiing to the body of Knowledge. Well done!! It is interesting well organized paper. It was pleasure to review it. Thank you for the opportunity.

6. PLOS authors have the option to publish the peer review history of their article (what does this mean?). If published, this will include your full peer review and any attached files.

---

## [Author Response · Author response to Decision Letter 0]

17 Jan 2023

We will like to thank the reviewers and the editor for the thorough review of this manuscript. It has helped improved the quality of the research. 

 Response: this has been checked and revised accordingly

2. We note that the grant information you provided in the ‘Funding Information’ and ‘Financial Disclosure’ sections do not match. When you resubmit, please ensure that you provide the correct grant numbers for the awards you received for your study in the ‘Funding Information’ section."

 Response: This has been corrected. No grant was received for the study

"MOF received funding to conduct the data collection and analysis"

Response: This has been corrected. The authors received no specific funding for this work

Response: This has been corrected. The authors received no specific funding for this work

Response: This has been corrected. The authors received no specific funding for this work

Response: This has been corrected. The authors received no specific funding for this work 

Response: this has been done

"The authors acknowledge afworo that provided funding support for the data collection and analysis of this piece of work."

Response: This has been corrected. The authors received no specific funding for this work

"MOF received funding to conduct the data collection and analysis"

Response: This has been corrected. The authors received no specific funding for this work 

Response: This has been corrected. The authors received no specific funding for this work 

Response: The data needed for this study can all be found in the study result section 

Response: The data needed for this study can all be found in the study result section 

Response: The data needed for this study can all be found in the study result section 

Response: The data needed for this study can all be found in the study result section 

Reviewer #1: • Add more papers from 2017 to 2022 in the introduction section. It should be motivated for the reader and also add unique contribution.

Response: A literature search was carried out prior to this resubmission and this elicited 4 new relevant studies that have been included in the introduction section

• Address theoretical background of the study and also add theoretical implication (Support with theories).

Response: Done. The Feminist institutionalism analysis approach was used for this study. See lines 102-110

• Author can mention objective of the study and research gap separately which enhance quality of paper.

Response: The final paragraphs have now been dedicated to a focused discussion relating back to the study objectives and research questions

• Add theoretical and managerial implications

 Response: Thank you for raising this useful point. Lines 425 to 442 expand on this issue

• Explain conclusion, limitation and future scope separately.

Response: We have included two paragraphs that discuss the strengths, limitations, and future scope of the study. See Lines 425 to 442. A paragraph has also been added to the conclusion section

• The language of the paper needs a careful editing.

 Response: The entire paper has been edited for style and language

• References must follow the style as per journal

 Response: The manuscript has been revised to adhere to the style of the journal

Reviewer #2: Paper looks good and I strongly believe that this paper is contributiing to the body of Knowledge. Well done!! It is interesting well organized paper. It was pleasure to review it. Thank you for the opportunity.

 Response: thank you for this positive feedback.

---

## [Decision Letter · Decision Letter 1]

16 Mar 2023

A qualitative insight into researchers' perceptions of gender inequality in medical and dental research institutions in Nigeria.

PONE-D-22-27154R1

Dear Dr. Folayan,

We’re pleased to inform you that your manuscript has been judged scientifically suitable for publication and will be formally accepted for publication once it meets all outstanding technical requirements.

Kind regards,

Claudia Noemi González Brambila, Ph.D.

Academic Editor

PLOS ONE

Additional Editor Comments (optional):

Reviewers' comments:

Reviewer's Responses to Questions

**Comments to the Author**

1. If the authors have adequately addressed your comments raised in a previous round of review and you feel that this manuscript is now acceptable for publication, you may indicate that here to bypass the “Comments to the Author” section, enter your conflict of interest statement in the “Confidential to Editor” section, and submit your "Accept" recommendation.

Reviewer #1: (No Response)

Reviewer #2: All comments have been addressed

2. Is the manuscript technically sound, and do the data support the conclusions?

Reviewer #1: Yes

Reviewer #2: Yes

3. Has the statistical analysis been performed appropriately and rigorously? 

Reviewer #1: Yes

Reviewer #2: N/A

4. Have the authors made all data underlying the findings in their manuscript fully available?

Reviewer #1: Yes

Reviewer #2: Yes

5. Is the manuscript presented in an intelligible fashion and written in standard English?

Reviewer #1: Yes

Reviewer #2: Yes

6. Review Comments to the Author

Reviewer #1: Good work. I can see you have incorporated all the comments. We can go ahead with the publication now.

Reviewer #2: (No Response)

7. PLOS authors have the option to publish the peer review history of their article (what does this mean?). If published, this will include your full peer review and any attached files.

Reviewer #1: **Yes: **Nishant Agrawal

Reviewer #2: **Yes: **BIBHAV ADHIKARI

---

## [Editor Report · Acceptance letter]

28 Mar 2023

PONE-D-22-27154R1 

A qualitative insight into researchers’ perceptions of gender inequality in medical and dental research institutions in Nigeria. 

Dear Dr. Folayan:

I'm pleased to inform you that your manuscript has been deemed suitable for publication in PLOS ONE. Congratulations! Your manuscript is now with our production department. 

Kind regards, 

on behalf of

Dr. Claudia Noemi González Brambila 

Academic Editor

PLOS ONE